# Learnable Polyphase Sampling for Shift Invariant and Equivariant Convolutional Networks

**Renan A. Rojas-Gomez**[*]     **Teck-Yian Lim**[*]
**Alexander G. Schwing**     **Minh N. Do**     **Raymond A. Yeh**[†]

Department of Electrical Engineering, University of Illinois at Urbana-Champaign
[†]Department of Computer Science, Purdue University

## Abstract

We propose learnable polyphase sampling (LPS), a pair of learnable down/up-sampling layers that enable truly shift-invariant and equivariant convolutional networks. LPS can be trained end-to-end from data and generalizes existing hand-crafted downsampling layers. It is widely applicable as it can be integrated into any convolutional network by replacing down/upsampling layers. We evaluate LPS on image classification and semantic segmentation. Experiments show that LPS is on-par with or outperforms existing methods in both performance and shift consistency. For the first time, we achieve true shift-equivariance on semantic segmentation (PASCAL VOC), *i.e.*, 100% shift consistency, outperforming baselines by an absolute 3.3%. Our project page and code are available at
https://raymondyeh07.github.io/learnable_polyphase_sampling/

## 1   Introduction

For tasks like image classification, shifts of an object do not change the corresponding object label, *i.e.*, the task is shift-invariant. This shift-invariance property has been incorporated into deep-nets yielding convolutional neural nets (CNN). Seminal works on CNNs [15, 24] directly attribute the model design to shift-invariance. For example, Fukushima [15] states "the network has an ability of position-invariant pattern recognition" and LeCun et al. [24] motivate CNNs by stating that they "ensure some degree of shift invariance."

CNNs have evolved since their conception. Modern deep-nets contain more layers, use different non-linearities and pooling layers. Re-examining these modern architectures, Zhang [56] surprisingly finds that modern deep-nets are not shift-invariant. To address this, Zhang [56] and Zou et al. [57] propose to perform anti-aliasing before each downsampling layer, and found it to improve the degree of invariance. More recently, Chaman and Dokmanic [5] show that deep-nets can be "truly shift-invariant," *i.e.*, a model's output is identical for given shifted inputs. For this, they replace all downsampling layers with their adaptive polyphase sampling (APS) layer.

While APS achieves true shift-invariance by selecting the max-norm polyphase component (a hand-crafted downsampling scheme), an important question arises: are there more effective downsampling schemes that can achieve true shift-invariance? Consider an extreme case, a handcrafted deep-net that always outputs zeros is truly shift-invariant, but does not accomplish any task. This motivates to study how truly shift-invariant downsampling schemes can be learned from data.

For this we propose Learnable Polyphase Sampling (LPS), a pair of down/upsampling layers that yield truly shift-invariant/equivariant deep-nets and can be trained in an end-to-end manner. For

---

[*]Equal contribution.

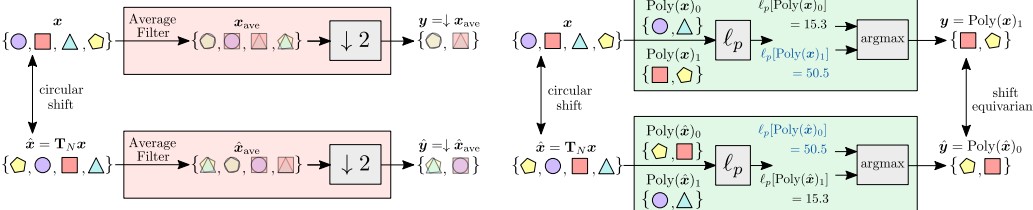

**Figure 1. Left: lowpass filtering (LPF)** [56] averages the inputs before downsampling to encourage shift-equivariance. The outputs $y$ and $\hat{y}$ are "more similar" due to the average filter. Despite improving shift consistency, the model is still shift variant by definition. **Right: adaptive polyphase sampling (APS)** [5] selects the component with the max $\ell_p$-norm. This hand-crafted rule is shift equivariant as it selects the square and pentagon in both $\hat{y}$ and $y$.

downsampling, LPS can be easily integrated into existing deep-net architectures by swapping out the pooling/striding layers. Theoretically, LPS generalizes APS to downsampling schemes that cannot be represented by APS. Hence, LPS's ideal performance is never worse than that of APS. For upsampling, LPS guarantees architectures that are truly shift-equivariant, *i.e.*, the output shifts accordingly when the input shifts. This is desirable for tasks like semantic image segmentation.

To validate the proposed LPS, we conduct extensive experiments: (a) image classification on CI-FAR10 [21] and ImageNet [12]; (b) semantic segmentation on PASCAL VOC [13]. We observe that the proposed approach outperforms APS and further improves anti-aliasing methods on both model performance and shift consistency. **Our contributions are as follows:**

- We propose learnable polyphase sampling (LPS), a pair of novel down/upsampling layers, and prove that they yield truly shift-invariant/-equivariant deep-nets. Different from prior works, our sampling scheme is trained from end-to-end and not handcrafted.
- We theoretically prove that LPS (downsampling) is a generalization of APS. Hence, in theory, LPS improves upon APS.
- We conduct extensive experiments demonstrating the effectiveness of LPS on image classification and segmentation over three datasets comparing to APS and anti-aliasing approaches.

## 2   Related Work

In this section, we briefly discuss related work, including shift invariant/equivariant deep-nets and pooling layers. Additional necessary concepts are reviewed in Sec. 3.

**Shift Invariant/equivariant convolutional networks.** Modern convolutional networks use striding or pooling to reduce the amount of memory and computation in the model [17, 22, 40, 45]. As pointed out by Azulay and Weiss [1] and Zhang [56], these pooling/striding layers break the shift-invariance property of deep-nets. To address this issue, Zhang [56] proposed to perform anti-aliasing, *i.e.*, lowpass filtering (LPF) before each downsampling, a canonical signal processing technique for multi-rate systems [47]. We illustrate this approach in Fig. 1 (left). Zou et al. [57] further improved the LPF technique by using adaptive filters which better preserve edge information.

While anti-aliasing filters are effective, Chaman and Dokmanic [5] show that true shift-invariance, *i.e.*, 100% shift consistency, can be achieved without anti-aliasing. Specifically, they propose Adaptive Polyphase Sampling (APS) which selects the downsampling indices, *i.e.*, polyphase components, based on the $\ell_p$-norm of the polyphase components; a handcrafted rule, as illustrated in Fig. 1 (right). In a follow up technical report [4], APS is extended to upsampling using unpooling layers [2, 55], where the downsampling indices are saved to place values back to their corresponding spatial location during upsampling. Our work presents a novel pair of shift-invariant/equivariant down/upsampling layers which are trainable, in contrast to APS's handcrafted selection rule.

We note that generalizations of equivariance beyond shifts have also been studied [3, 7, 37, 39, 43, 46, 48, 52] and applied to various domains, *e.g.*, sets [16, 31, 35, 38, 50, 53], graphs [10, 11, 19, 27, 28, 30, 32, 44, 51], spherical images [8, 9, 20], volumetric data [49], *etc*. In this work, we focus solely on shift-equivariance for images with CNNs.

**Pooling layers.** Many designs for better downsampling or pooling layers have been proposed. Popular choices are Average-Pooling [23] and Max-Pooling [36]. Other generalizations also exists, *e.g.*, $L_P$-

Pooling [42] which generalizes pooling to use different norms. The effectiveness of different pooling layers has also been studied by Scherer et al. [41]. More similar to our work is Stochastic-Pooling [54] and Mixed Max-Average Pooling [25]. Stochastic-Pooling constructs a probability distribution by normalizing activations within a window and sampling during training. In our work, we present a novel design which learns the sampling distribution. Mixed Max-Average Pooling learns a single scalar to permit a soft-choice between Max- and Average-Pooling. In contrast, our LPS has shift-equivariance guarantees while being end-to-end trainable.

# 3 Preliminaries

We provide a brief review on equivariant and invariant functions to establish the notation. For readability, we use one-dimensional data to illustrate these ideas. In practice, these concepts are generalized to multiple channels and two-dimensional data.

**Shift invariance and equivariance.** The concept of *equivariance*, a generalization of invariance, describes how a function's output is transformed given that the input is transformed in a *predefined way*. For example, shift equivariance describes how the output is shifted given that the input is also shifted: think of image segmentation, if an object in the image is shifted then its corresponding mask is also shifted.

A function $f : \mathbb{R}^N \mapsto \mathbb{R}^M$ is $\boldsymbol{T}_N, \{\boldsymbol{T}_M, \boldsymbol{I}\}$-*equivariant* (shift-equivariant) if and only if *(iff)*

$$\exists \, T \in \{\boldsymbol{T}_M, \boldsymbol{I}\} \text{ s.t. } f(\boldsymbol{T}_N \boldsymbol{x}) = T f(\boldsymbol{x}) \, \forall \boldsymbol{x} \in \mathbb{R}^N, \tag{1}$$

where $\boldsymbol{T}_N \boldsymbol{x}[n] \triangleq \boldsymbol{x}[(n+1) \bmod N] \, \forall n \in \mathbb{Z}$ denotes a circular shift, $[\cdot]$ denotes the indexing operator, and $\boldsymbol{I}$ denotes the identity function. This definition of equivariance handles the ambiguity that arises when shifting by one and downsampling by two. Ideally, a shift by one at the input should result in a 0.5 shift in the downsampled signal, which is not achievable on the integer grid. Hence, this definition considers *either a shift by one or a no shift* at the output as equivariant.

Following the equivariance definition, invariance can be viewed as a special case where the transformation at the output is an identity function, $\boldsymbol{I}$. Concretely, a function $f : \mathbb{R}^N \mapsto \mathbb{R}^M$ is $\boldsymbol{T}_N, \{\boldsymbol{I}\}$-equivariant (shift-invariant) *iff*

$$f(\boldsymbol{T}_N \boldsymbol{x}) = f(\boldsymbol{x}) \, \forall \boldsymbol{x} \in \mathbb{R}^N. \tag{2}$$

To obtain shift-invariance from shift-equivariant functions it is common to use global pooling. Observe that

$$\sum_m f(\boldsymbol{T} \boldsymbol{x})[m] = \sum_m (\boldsymbol{T} f(\boldsymbol{x}))[m] \tag{3}$$

is shift-invariant if $f$ is shift-equivariant, as summation is an orderless operation. Note that the composition of shift-equivariant functions maintains shift-equivariance. Hence, $f$ can be a stack of equivariant layers, *e.g.*, a composition of convolution layers.

While existing deep-nets [17, 26, 40] do use global spatial pooling, these architectures are *not* shift-invariant. This is due to pooling and downsampling layers, which are not shift-equivariant as we review next.

**Downsampling and pooling layers.** A downsampling-by-two layer $\mathrm{D} : \mathbb{R}^N \mapsto \mathbb{R}^{\lfloor N/2 \rfloor}$ is defined as

$$\mathrm{D}(\boldsymbol{x})[n] = \boldsymbol{x}[2n] \, \forall n \in \mathbb{Z}, \tag{4}$$

which returns the even indices of the input $\boldsymbol{x}$. As a shift operator makes the odd indices even, a downsampling layer is not shift-equivariant/invariant.

Commonly used average or max pooling can be viewed as an average or max filter followed by downsampling, hence pooling is also not shift-equivariant/invariant. To address this issue, Chaman and Dokmanic [5] propose adaptive polyphase sampling (APS) which is an input dependent (adaptive) selection of the odd/even indices.

**Adaptive polyphase sampling.** Proposed by Chaman and Dokmanic [5], adaptive polyphase sampling (APS) returns whether the odd or even indices, *i.e.*, the polyphase components, based on their norms. Formally, APS : $\mathbb{R}^N \mapsto \mathbb{R}^{\lfloor N/2 \rfloor}$ is defined as:

$$\mathrm{APS}(\boldsymbol{x}) = \begin{cases} \mathrm{Poly}(\boldsymbol{x})_0 & \text{if } \|\mathrm{Poly}(\boldsymbol{x})_0\| > \|\mathrm{Poly}(\boldsymbol{x})_1\| \\ \mathrm{Poly}(\boldsymbol{x})_1 & \text{otherwise} \end{cases}, \tag{5}$$

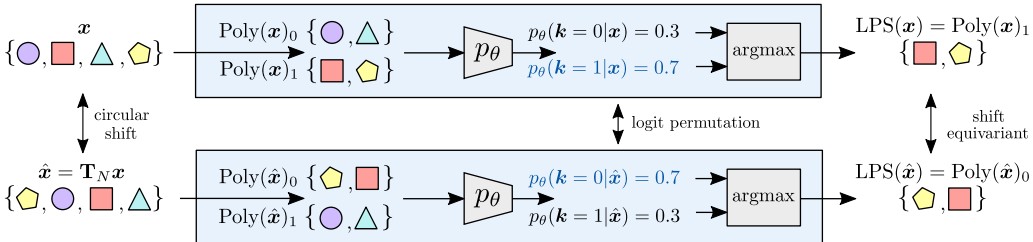

**Figure 2.** Illustration of our proposed LPD design. Inspired by the polyphase permutation property, *Lemma 1*, the likelihood of selecting a polyphase component is learned with a shift-permutation equivariant CNN model $p_\theta$. This enforces components between the input and its shifted version to have consistent logits, which leads to a shift-equivariant downsampling layer.

where $\boldsymbol{x} \in \mathbb{R}^N$ is the input and $\text{Poly}(\boldsymbol{x})_i$ denotes the polyphase components, *i.e.*,

$$\text{Poly}(\boldsymbol{x})_0[n] = \boldsymbol{x}[2n] \ \text{ and } \ \text{Poly}(\boldsymbol{x})_1[n] = \boldsymbol{x}[2n+1]. \tag{6}$$

While this handcrafted selection rule achieves a consistent selection of the polyphase components, it is *not the only way* to achieve it, *e.g.*, returning the polyphase component with the smaller norm. In this work, we study a family of shift-equivariant sampling layers and propose how to learn them in a data-driven manner.

## 4 Approach

Our goal is to design a learnable down/upsampling layer that is shift-invariant/equivariant. We formulate down/upsampling by modeling the conditional probability of selecting each polyphase component given an input. For this we use a small neural network. This enables the sampling scheme to be trained end-to-end from data, hence the name learnable polyphase sampling (LPS).

In Sec. 4.1, we introduce learnable polyphase downsampling (LPD), discuss how to train it end-to-end, and show that it generalizes APS. In Sec. 4.2, we propose a practical layer design of LPD. Lastly, in Sec. 4.3, we discuss how to perform LPS for upsampling, namely, learnable polyphase upsampling (LPU). For readability, we present the approach using one dimensional data, *i.e.*, a row in an image.

### 4.1 Learnable Polyphase Downsampling

We propose learnable polyphase downsampling (LPD) to learn a shift-equivariant downsampling layer. Given an input feature map $\boldsymbol{x} \in \mathbb{R}^{C \times N}$, LPD spatially downsamples the input to produce an output in $\mathbb{R}^{C \times \lfloor N/2 \rfloor}$ via

$$\text{LPD}(\boldsymbol{x})[c, n] = \boldsymbol{x}[c, 2n + k^\star] \triangleq \text{Poly}(\boldsymbol{x})_{k^\star}, \tag{7}$$

where $k^\star = \arg\max_{k \in \{0,1\}} p_\theta(\mathbf{k} = k|\boldsymbol{x})$ and $\text{Poly}(\boldsymbol{x})_{k^\star}$ denotes the $k^\star$-th polyphase component. We model a conditional probability $p_\theta(\mathbf{k}|\boldsymbol{x})$ for selecting polyphase components, *i.e.*, $\mathbf{k}$ denotes the random variable of the polyphase indices. For 1D data, there are only two polyphase components.

Critically, *not all* $p_\theta$ lead to an equivariant downsampling layer. For example, $p_\theta(\mathbf{k} = 0|\boldsymbol{x}) = 1$ results in the standard down-sampling which always returns values on even indices for 1D signals. We will next examine which family of $p_\theta$ achieves a shift-equivariant downsampling layer.

**Shift-permutation equivariance of** $p_\theta$**.** Consider the example in Fig. 2. We can see that a circular shift in the spatial domain induces a permutation in the polyphase components. Observe that the top-row of the polyphase component containing the blue circle and orange square are permuted to the second row when the input is circularly shifted. We now state this formally.

> **Lemma 1.** *Polyphase shift-permutation property*
>
> $$Poly(\boldsymbol{T}_N \boldsymbol{x})_k = \begin{cases} Poly(\boldsymbol{x})_1 & if \ k = 0 \\ \boldsymbol{T}_M Poly(\boldsymbol{x})_0 & if \ k = 1 \end{cases}. \tag{8}$$

*Proof.* By definition, $\text{Poly}(\boldsymbol{T}_N\boldsymbol{x})_k[n]$

$$= \boldsymbol{T}_N\boldsymbol{x}[(2n+k) \bmod N] = \boldsymbol{x}[(2n+k+1) \bmod N] \tag{9}$$

$$= \begin{cases} \boldsymbol{x}[(2n+1) \bmod N] &= \text{Poly}(\boldsymbol{x})_1 & \text{if } k = 0 \\ \boldsymbol{x}[(2(n+1)) \bmod N] = \boldsymbol{T}_M\text{Poly}(\boldsymbol{x})_0 & \text{if } k = 1 \end{cases} \tag{10}$$

$\square$

From Lemma 1, we observe that to achieve an equivariant downsampling layer a spatially **shifted input should lead to a permutation of the selection probability** (Claim 1). We note that $p_\theta$ is said to be *shift-permutation-equivariant* if

$$p_\theta(\mathbf{k} = \pi(k)|\boldsymbol{T}_N\boldsymbol{x}) = p_\theta(\mathbf{k} = k|\boldsymbol{x}), \tag{11}$$

where $\pi$ denotes a permutation on the polyphase indices, *i.e.*, a "swap" of indices is characterized by $\pi(k)$, *i.e.*, $\pi(0) = 1$ and $\pi(1) = 0$.

> **Claim 1.** *If $p_\theta$ is shift-permutation-equivariant, defined in Eq. (11), then LPD defined in Eq. (7) is a shift-equivariant downsampling layer.*

*Proof.* Let $\hat{\boldsymbol{x}} \triangleq \boldsymbol{T}_N\boldsymbol{x}$ be a shifted version of $\boldsymbol{x} \in \mathbb{R}^N$. Recall $\text{LPD}(\boldsymbol{x})$ and $\text{LPD}(\hat{\boldsymbol{x}})$ are defined as:

$$\text{LPD}(\boldsymbol{x}) \triangleq \text{Poly}(\boldsymbol{x})_{k^\star}, \ k^\star = \underset{k \in \{0,1\}}{\arg\max} \ p_\theta(\mathbf{k} = k|\boldsymbol{x}), \tag{12}$$

$$\text{LPD}(\hat{\boldsymbol{x}}) \triangleq \text{Poly}(\hat{\boldsymbol{x}})_{\hat{k}^\star}, \ \hat{k}^\star = \underset{k \in \{0,1\}}{\arg\max} \ p_\theta(\mathbf{k} = k|\hat{\boldsymbol{x}}). \tag{13}$$

From Lemma 1, $\text{LPD}(\boldsymbol{T}_N\boldsymbol{x})$ can be expressed as:

$$\text{LPD}(\boldsymbol{T}_N\boldsymbol{x}) = \begin{cases} \text{Poly}(\boldsymbol{x})_1 & \text{if } \hat{k}^\star = 0 \\ \boldsymbol{T}_M\text{Poly}(\boldsymbol{x})_0 & \text{if } \hat{k}^\star = 1 \end{cases}. \tag{14}$$

As $p_\theta$ is the shift-permutation-equivariant,

$$\hat{k}^\star = \pi(k^\star) = 1 - k^\star. \tag{15}$$

Finally, combining Eq. (14) and Eq. (15),

$$\text{LPD}(\boldsymbol{T}_N\boldsymbol{x}) = \begin{cases} \text{Poly}(\boldsymbol{x})_1 & \text{if } k^\star = 1 \\ \boldsymbol{T}_M\text{Poly}(\boldsymbol{x})_0 & \text{if } k^\star = 0 \end{cases} = \big((1 - k^\star)\boldsymbol{T}_M + k^\star\boldsymbol{I}\big) \cdot \text{LPD}(\boldsymbol{x}), \tag{16}$$

showing that LPD satisfies the shift-equivariance definition reviewed in Eq. (1). $\square$

Here, we parameterize $p_\theta$ with a small neural network. The exact construction of a shift-permutation equivariant deep-net architecture is deferred to Sec. 4.2. We next discuss how to train the distribution parameters $\theta$ in LPD.

**End-to-end training of LPD.** At training time, to incorporate stochasticity and compute gradients, we parameterize $p_\theta$ using Gumbel Softmax [18, 29]. To backpropagate gradients to $\theta$, we relax the selection of polyphase components as a convex combination, *i.e.*,

$$\boldsymbol{y} = \sum_k \boldsymbol{z}_k \cdot \text{Poly}(\boldsymbol{x})_k, \ \ \boldsymbol{z} \sim p_\theta(\boldsymbol{k}|\boldsymbol{x}), \tag{17}$$

where $\boldsymbol{z}$ corresponds to a selection variable, *i.e.*, $\sum_k \boldsymbol{z}_k = 1$ and $\boldsymbol{z}_k \in [0, 1]$. Note the slight abuse of notation as $p_\theta(\boldsymbol{k}|\boldsymbol{x})$ denotes a probability over polyphase indices represented in a one-hot format. We further encourage the Gumbel Softmax to behave more like an argmax by decaying its temperature $\tau$ during training as recommended by Jang et al. [18].

**LPD generalizes APS.** A key advantage of LPS over APS is that it can learn from data, potentially leading to a better sampling scheme than a handcrafted one. Here, we show that APS is a special case of LPD. Therefore, LPD should perform at least as well as APS if parameters are trained well.

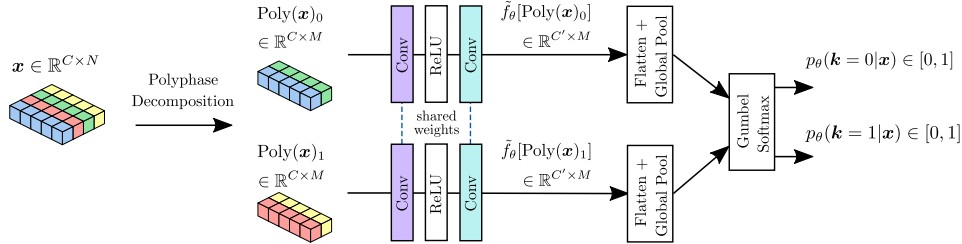

**Figure 3.** Proposed shift-permutation equivariant model. By sharing weights across components and computing logits via global pooling, this neural network is shift-permutation equivariant. To introduce stochasticity during training, we adopt the Gumbel Softmax for sampling.

**Claim 2.** *APS is a special case of LPD,* i.e., *LPD can represent APS's selection rule.*

*Proof.* Consider a parametrization of $p_\theta$ as follows,

$$p_\theta(\mathbf{k} = k|\boldsymbol{x}) = \frac{\exp\left(\|\text{Poly}(\boldsymbol{x})_k\|\right)}{\sum_j \exp(\|\text{Poly}(\boldsymbol{x})_j\|)}. \tag{18}$$

As the exponential is a strictly increasing function we have

$$\arg\max_k p_\theta(\mathbf{k} = k|\boldsymbol{x}) = \arg\max_k \|\text{Poly}(x)_k\|. \tag{19}$$

Eq. (18) is a softmax with input $\|\text{Poly}(\boldsymbol{x})_k\|$, as such a function exists, LPD generalizes APS. $\square$

## 4.2 Practical LPD Design

We aim for a conditional distribution $p_\theta$ that is shift-permutation equivariant to obtain a shift-equivariant pooling layer. Let the conditional probability be modeled as:

$$p_\theta(\mathbf{k} = k|\boldsymbol{x}) \triangleq \frac{\exp[f_\theta(\text{Poly}(\boldsymbol{x})_k)]}{\sum_j \exp[f_\theta(\text{Poly}(\boldsymbol{x})_j)]}, \tag{20}$$

where $f_\theta : \mathbb{R}^{C \times H' \times W'} \mapsto \mathbb{R}$ is a small network that extracts features from polyphase component $\text{Poly}(\boldsymbol{x})_k$. We first show that $p_\theta$ is shift-permutation equivariant if $f_\theta$ is shift invariant.

**Claim 3.** *In Eq. (20), if $f_\theta$ is shift invariant then $p_\theta$ is shift-permutation equivariant (Eq. (11)).*

*Proof.* Denote a feature map $\boldsymbol{x}$ and its shifted version $\hat{\boldsymbol{x}} \triangleq \boldsymbol{T}_N \boldsymbol{x}$. By definition,

$$p_\theta(\mathbf{k} = \pi(k)|\boldsymbol{T}_N\boldsymbol{x}) = \frac{\exp(f_\theta(\text{Poly}(\boldsymbol{T}_N\boldsymbol{x})_{\pi(k)}))}{\sum_j \exp(f_\theta(\text{Poly}(\boldsymbol{T}_N\boldsymbol{x})_j))}. \tag{21}$$

With a shift-invariant $f_\theta$ and using Lemma 1,

$$f_\theta(\text{Poly}(\boldsymbol{T}_N\boldsymbol{x})_{\pi(k)}) = f_\theta(\boldsymbol{T}_M\text{Poly}(\boldsymbol{x})_k) = f_\theta(\text{Poly}(\boldsymbol{x})_k) \tag{22}$$

$$\therefore\ p_\theta(\boldsymbol{k} = \pi(k)|\boldsymbol{T}_N\boldsymbol{x}) = \frac{\exp(\text{Poly}(\boldsymbol{x})_k)}{\sum_j \exp(\mathbf{k} = \text{Poly}(\boldsymbol{x})_j)} = p_\theta(\mathbf{k} = k|\boldsymbol{x}) \qquad \square$$

Based on the result in Claim 3, we now present a convolution based meta-architecture that satisfies the shift-permutation property. The general design principle: share parameters across polyphase indices, just as convolution achieves shift equivariance by sharing parameters, plus averaging over the spatial domain. An illustration of the proposed meta-architecture is shown in Fig. 3.

**Fully convolutional model.** Logits are extracted from the polyphase components via fully-convolutional operations followed by averaging along the channel and the spatial domain. Following this, $f_\theta^{\text{conv}}$ is denoted as:

$$f_\theta^{\text{conv}}(\text{Poly}(\boldsymbol{x})_k) \triangleq \frac{1}{CM} \sum_{c,n} \tilde{f}_\theta^{\text{conv}}(\text{Poly}(\boldsymbol{x})_k)[c, n], \tag{23}$$

where $\tilde{f}_\theta^{\text{conv}} : \mathbb{R}^{C \times M} \mapsto \mathbb{R}^{C \times M}$ is a CNN model (without pooling layers) and $M = \lfloor N/2 \rfloor$. The shift equivariance property of $\tilde{f}_\theta^{\text{conv}}$ guarantees that $f_\theta^{\text{conv}}$ is shift-invariant due to the global pooling.

### 4.3 Learnable Polyphase Upsampling (LPU)

Beyond shift invariant models, we extend the theory from downsampling to upsampling, which permits to design shift-equivariant models. The main idea is to place the features obtained after downsampling back to their original spatial location. Given a feature map $\boldsymbol{y} \in \mathbb{R}^{C \times \lfloor N/2 \rfloor}$ downsampled via LPD from $\boldsymbol{x}$, the upsampling layer outputs $\boldsymbol{u} \in \mathbb{R}^{C \times N}$ are defined as follows:

$$\text{Poly}(\boldsymbol{u})_{k^\star} = \begin{cases} \boldsymbol{y}, & k^\star = \arg\max_{k \in \{0,1\}} p_\theta(\boldsymbol{k} = k | \boldsymbol{x}) \\ \boldsymbol{0}, & \text{otherwise.} \end{cases} \tag{24}$$

We name this layer learnable polyphase upsampling (LPU), $i.e.$, $\text{LPU}(\boldsymbol{y}, p_\theta) \triangleq \boldsymbol{u}$. We now show that LPU and LPD achieve shift-equivariance.

> **Claim 4.** *If $p_\theta$ is shift-permutation equivariant, as defined in Eq. (11), then LPU ∘ LPD is shift-equivariant.*

*Proof.* We prove this claim following definitions of LPU, LPD and Lemma 1. The complete proof is deferred to Appendix Sec. A1. □

**End-to-end training of LPU.** As in downsampling, we also incorporate stochasticity via Gumbel-Softmax. To backpropagate gradients to $p_\theta$, we relax the hard selection into a convex combination, $i.e.$,

$$\text{Poly}(\boldsymbol{u})_k = \boldsymbol{z}_k \cdot \boldsymbol{y}, \quad \boldsymbol{z} \sim p_\theta(\boldsymbol{k} | \boldsymbol{x}). \tag{25}$$

**Anti-aliasing for upsampling.** While LPU provides a shift-equivariant upsampling scheme, it introduces zeros in the output which results in high-frequency components. This is known as aliasing in a multirate system [47]. To resolve this, following the classical solution, we apply a low-pass filter scaled by the upsampling factor after each LPU.

## 5 Experiments

We conduct experiments on image classification following prior works. We report on the same architectures and training setup. We report both the circular shift setup in APS [5] and the standard shift setup in LPF [56].

We also evaluate on semantic segmentation, considering the circular shift, inspired by APS, and the standard shift setup following DDAC [57]. For circular shift settings, the theory exactly matches the experiment hence true equivariance is achieved. To our knowledge, this is the first truly shift equivariant model reported on PASCAL VOC.

**Table 1.** ResNet-18 (CIFAR10) top-1 accuracy and circular shift consistency. LPS outperforms all alternative pooling and anti-aliasing methods.

| Method | Anti-Alias | Acc. ↑ | C-Cons. ↑ |
|---|---|---|---|
| Baseline | - | $91.44 \pm 0.33$ | $89.64 \pm 0.39$ |
| APS | - | $94.07 \pm 0.28$ | $\mathbf{100 \pm 0.0}$ |
| LPS (Ours) | - | $\mathbf{94.45 \pm 0.05}$ | $\mathbf{100 \pm 0.0}$ |
| LPF | Rect-2 | $93.1 \pm 0.17$ | $94.75 \pm 0.43$ |
| APS | Rect-2 | $94.38 \pm 0.25$ | $\mathbf{100 \pm 0.0}$ |
| LPS (Ours) | Rect-2 | $\mathbf{94.69 \pm 0.06}$ | $\mathbf{100 \pm 0.0}$ |
| LPF | Tri-3 | $93.97 \pm 0.18$ | $96.74 \pm 0.2$ |
| APS | Tri-3 | $94.36 \pm 0.17$ | $\mathbf{100 \pm 0.0}$ |
| LPS (Ours) | Tri-3 | $\mathbf{94.8 \pm 0.14}$ | $\mathbf{100 \pm 0.0}$ |
| LPF | Bin-5 | $94.43 \pm 0.15$ | $98.34 \pm 0.15$ |
| APS | Bin-5 | $94.44 \pm 0.19$ | $\mathbf{100 \pm 0.0}$ |
| LPS (Ours) | Bin-5 | $\mathbf{94.49 \pm 0.1}$ | $\mathbf{100 \pm 0.0}$ |
| DDAC | Adapt-3 | $93.17 \pm 0.19$ | $95.13 \pm 0.15$ |
| APS | Adapt-3 | $94.42 \pm 0.13$ | $\mathbf{100 \pm 0.0}$ |
| LPS (Ours) | Adapt-3 | $\mathbf{94.57 \pm 0.12}$ | $\mathbf{100 \pm 0.0}$ |

### 5.1 Image Classification (Circular Shift)

**Experiment & implementation details.**

Following APS, all the evaluated pooling and anti-aliasing models use the ResNet-18 [17] architecture with circular padding on CIFAR10 [21] and ImageNet [12]. Anti-alias filters are applied after each downsampling layer following LPF [56] and DDAC [57]. We also replace downsampling layers with APS [5] and our proposed LPS layer. We provide more experimental details in Appendix Sec. A4.

**Evaluation metrics.** We report classification accuracy to quantify the model performance on the original dataset without any shifts. To evaluate shift-invariance, following APS, we report circular-consistency (C-Cons.) which computes the average percentage of predicted labels that are equal under two different circular shifts, $i.e.$,

$$\hat{\boldsymbol{y}}(\text{Circ. Shift}_{h1,w1}(I)) = \hat{\boldsymbol{y}}(\text{Circ. Shift}_{h2,w2}(I)), \tag{26}$$

where $\hat{y}(I)$ denotes the predicted label for an input image $I$ and $h1, w1, h2, w2$ are uniformly sampled from 0 to 32. We report the average over five random seeds.

**CIFAR10 results.** Tab. 1 shows the classification accuracy and circular consistency on CIFAR10. We report the mean and standard deviation over five runs with different random initialization of the ResNet-18 model. We observe that the proposed LPS improves classification accuracy over all baselines while achieving 100% circular consistency. In addition to attaining perfect shift consistency, we observe that combining anti-aliasing with LPS further improves performance.

**ImageNet results.** We conduct experiments on ImageNet with circular shift using ResNet-18. In Tab. 2, we compare with APS's best model using a box filter (Rectangle-2), as reported by Chaman and Dokmanic [5]. While both APS and LPS achieve 100% circular consistency, our proposed LPS improves on classification accuracy in all scenarios, highlighting its advantages.

**Table 2.** ResNet-18 top-1 classification accuracy and circular consistency on ImageNet.

| Method | Anti-Alias | Acc. ↑ | C-Cons. ↑ |
|---|---|---|---|
| Baseline | - | 64.88 | 80.39 |
| APS | - | 67.05 | **100** |
| LPS (Ours) | - | **67.39** | **100** |
| LPF | Rect-2 | 67.03 | 84.35 |
| APS | Rect-2 | 67.6 | **100** |
| LPS (Ours) | Rect-2 | **68.45** | **100** |
| DDAC | Adapt-3 | 67.6 | 77.23 |
| APS | Adapt-3 | 69.02 | **100** |
| LPS (Ours) | Adapt-3 | **69.11** | **100** |

**Table 3.** ResNet-50/101 top-1 classification accuracy and shift consistency on ImageNet.

| Method | Anti-Alias | Model | Acc. ↑ | S-Cons. ↑ |
|---|---|---|---|---|
| Baseline | - | ResNet-50 | 76.16 | 89.2 |
| LPF | Tri-3 | ResNet-50 | 76.83 | 90.91 |
| LPS (Ours) | Tri-3 | ResNet-50 | **77.14** | **91.43** |
| Baseline | - | ResNet-101 | 77.7 | 90.6 |
| LPF | Tri-3 | ResNet-101 | 78.4 | 91.6 |
| LPS | Tri-3 | ResNet-101 | **78.51** | **91.69** |
| DDAC | Adapt-3 | ResNet-101 | **79.0** | 91.8 |
| DDAC* | Adapt-3 | ResNet-101 | 78.64 | 91.83 |
| LPS (Ours) | Adapt-3 | ResNet-101 | 78.8 | **92.4** |

## 5.2 Image Classification (Standard Shift)

**Experiment & implementation details.** To directly compare with results from LPF and DDAC, we conduct experiments on ImageNet using the ResNet-50 and ResNet-101 architectures following their setting, *i.e.*, training with standard shifts augmentation and using convolution layers with zero-padding.

**Evaluation metrics.** Shift consistency (S-Cons.) computes the average percentage of

$$\hat{y}(\text{Shift}_{h1,w1}(I)) = \hat{y}(\text{Shift}_{h2,w2}(I)), \tag{27}$$

where $h1, w1, h2, w2$ are uniformly sampled from the interval $\{0, \ldots, 32\}$. To avoid padding at the boundary, following LPF [56], we perform a shift on an image then crop its center $224 \times 224$ region. We note that, due to the change in content at the boundary, perfect shift consistency is not guaranteed.

**ImageNet results.** In Tab. 3, we compare to the best anti-aliasing result as reported in LPF, DDAC and DDAC* which is trained from the authors' released code using hyperparameters specified in the repository. Note, in standard shift setting LPS no longer achieves true shift-invariance due to padding at the boundaries. Despite this gap from the theory, LPS achieves improvements in both performance and shift-consistency over the baselines. When compared to LPF, both ResNet-50 and ResNet-101 architecture achieved improved classification accuracy and shift-consistency. When compared to DDAC, LPS achieves comparable accuracy with higher shift-consistency.

## 5.3 Trainable Parameters and Inference Time

While LPD is a data-driven downsampling layer, we show that the additional trainable parameters introduced by it are marginal with respect to the classification architecture. Tab. 4 shows the number of trainable parameters required by the ResNet-101 models.

For each method, we report the *absolute* number of trainable parameters, which includes both classifier and learnable pooling weights. We also include the *relative* number of trainable parameters, which only considers the learnable pooling weights and the percentage it represents with respect to the default ResNet-101 architecture weights.

For comparison purposes, we also include the inference time required by each model to evaluate their computational overhead. Mean and standard deviation of the inference time is computed for

**Table 4.** Number of trainable parameters and inference time required by ResNet-101 using our proposed LPD layer and alternative methods. *Absolute* corresponds to the total number of trainable parameters, including classifier and pooling layers. *Relative* corresponds to the parameters of the pooling layers only. Inference time statistics computed on 100 batches with 32 images of size $224 \times 224 \times 3$ each.

| Pooling | Anti-alias | Trainable Parameters | | Inference Time | |
|---|---|---|---|---|---|
| | | Absolute | Relative | Mean (ms) | Std (ms) |
| LPD | LPF (Tri-3) | $42,966,730$ | $446,080$ (1.05%) | 83.61 | 0.19 |
| LPD | DDAC (Adapt-3) | $44,751,034$ | $2,230,384$ (5.24%) | 130.69 | 0.2 |
| ResNet-101 (Default) | DDAC (Adapt-3) | $44,304,954$ | $1,784,304$ (4.2%) | 124.69 | 0.25 |
| APS | LPF (Tri-3) | $42,520,650$ | 0 | 77.68 | 0.22 |
| ResNet-101 (Default) | LPF (Tri-3) | $42,520,650$ | 0 | 68.72 | 0.4 |

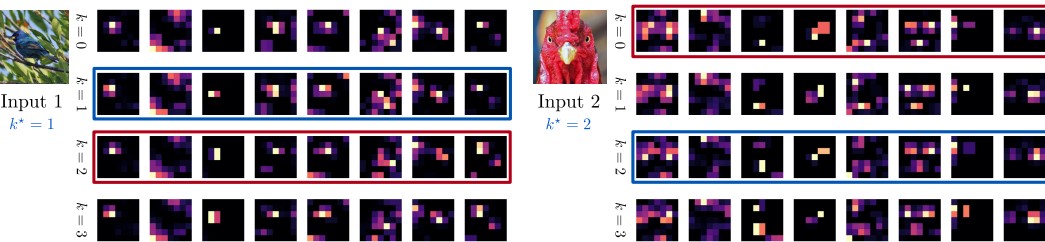

**Figure 4.** ResNet-50 LPS activation maps (ImageNet). We show all four polyphase components of the fourth layer (first 8 channels). The component selected by LPS is boxed in blue, while that with the largest $\ell_2$-norm is boxed in red. This indicates LPS did not learn to reproduce APS.

each method on 100 batches of size 32. Following ImageNet default settings, the image dimensions corrrespond to $224 \times 224 \times 3$.

Results show our proposed LPD method introduces approximately 1% additional trainable parameters on the ResNet-101 architecture, and increases the inference time roughly by 14.89 ms over the LPF anti-aliasing method (the less computationally expensive of the evaluated techniques). On the other hand, most of the overhead comes from DDAC, which increases the number of trainable parameters by approximately 4% and the inference time by approximately 55.97 ms. Overall, our comparison shows that, by equipping a classifier with LPD layers, the computational overhead is almost trivial.

Despite increasing the number of trainable parameters, we empirically show that our LPD approach outperforms classifiers with significantly more parameters. Please refer to Sec. A4.3 for additional experiments comparing the performance of our ResNet-101 + LPD model against the much larger ResNet-152 classifier.

**LPD learns sampling schemes different from APS.** To further analyze LPD, we replace all the LPD layers with APS for a ResNet-101 model trained on ImageNet. We observe a critical drop in top-1 classification accuracy from 78.8% to 0.1%, indicating that LPD did not learn a downsampling scheme equivalent to APS. We also counted how many times (across all layers) LPD selects the max-norm. On the ImageNet validation set, LPD selects the max $\ell_2$-norm polyphase component only 20.57% of the time. These show LPD learned a selection rule that differs from the handcrafted APS.

**Qualitative study on LPD.** In Fig. 4 we show the selected activations, at the fourth layer, of a ResNet-50 model with LPD. Each column describes the first 8 channels of the four possible polyphase components $k \in \{0, \ldots, 3\}$. The component selected by LPD, denoted as $k^\star$, is boxed in blue. For comparison purposes, we also boxed the component that maximizes the $\ell_2$-norm in red. We observe that LPD is distinct from APS as they select a different set of polyphase components. However, we did not observe a specific pattern that can explain LPD's selection rule.

### 5.4 Semantic Segmentation (Circular Shift)

**Experiment & implementation details.** We evaluate LPS's down/upsampling layers on semantic segmentation. As in DDAC [57], we evaluate using the PASCAL VOC [13] dataset. Following DDAC, we use DeepLabV3+ [6] as our baseline model. We use the ResNet-18 backbone pre-trained on the ImageNet (circular shift) reported in Sec. 5.1. We experiment with using only the LPD backbone and the full LPS, *i.e.*, both LPD and LPU.

**Table 5.** Semantic segmentation with circular shifts (ResNet-18 with DeepLabV3+) on PASCAL VOC.

| Method | Anti-Alias | mIoU↑ | mASCC↑ |
|--------|------------|-------|--------|
| DeepLabV3+ | - | 70.03 | 95.42 |
| LPF | Rect-2 | 71.02 | 96.03 |
| DDAC | Adapt-3 | 72.28 | 95.98 |
| APS | Adapt-3 | 72.37 | 96.70 |
| LPD only | Adapt-3 | **72.47** | 96.23 |
| LPS (Ours) | Adapt-3 | 72.37 | **100** |

**Table 6.** Semantic segmentation with standard shifts (ResNet-101 with DeepLabV3+) on PASCAL VOC. Results above the line are from DDAC's paper.

| Method | Anti-Alias | mIoU ↑ | mASSC ↑ |
|--------|------------|--------|---------|
| DeepLabV3+ | - | 78.5 | 95.5 |
| LPF | Tri-3 | 79.4 | 95.9 |
| DDAC | Adapt-3 | 80.3 | 96.3 |
| DDAC* | Adapt-3 | 80.31 | 97.83 |
| LPS (Ours) | Adapt-3 | **80.43** | **97.98** |

We also evaluate the performance using APS, which corresponds to a hand-crafted downsampling scheme, in combination with the default bilinear interpolation strategy from DeepLabV3+. Note that, while our LPS approach consists of both shift equivariant down and upsampling schemes (LPD and LPU, respectively), APS only operates on the downsampling process. Thus, the latter does not guarantee a circularly shift equivariant segmentation.

**Evaluation metric.** We report mean intersection over union (mIoU) to evaluate segmentation performance. To quantify circular-equivariance, we report mean Average Segmentation Circular Consistency (mASCC) which computes the average percentage of predicted (per-pixel) labels that remained the same under two different circular shifts. *I.e.*, a shifted image is passed to a model to make a segmentation prediction. This prediction is then "unshifted" for comparison. We report five random shift pairs for each image.

**Results.** We report the results for PASCAL VOC in Tab. 5. Overall, we observe that LPD only and LPS achieve comparable results to DDAC and APS in mIoU. Notably, LPS achieves 100% mASCC, matching the theory. This confirms that both the proposed LPD and LPU layers are necessary and are able to learn effective down/up sampling schemes for semantic segmentation.

### 5.5 Semantic Segmentation (Standard Shift)

**Experiment & implementation details.** For the standard shift setting, we directly follow the experimental setup from DDAC. We use DeepLabV3+ with a ResNet-101 backbone pre-trained on ImageNet as reported in Sec. 5.2.

**Evaluation metric.** To quantify the shift-equivariance, following DDAC, we report the mean Average Semantic Segmentation Consistency (mASSC) which is a linear-shift version of mASCC described in Sec. 5.4 except boundary pixels are ignored.

**Results.** In Tab. 6, we compare mIoU and mASSC of LPS to various baselines. We observe that LPS achieves improvements in mIoU and consistency when compared to DDAC*. We note that DDAC [57] did not release their code for mASSC. For a fair comparison, we report the performance of their released checkpoint using our implementation of mASSC, indicated with DDAC*. Despite the gap in theory and practice due to non-circular padding at the boundary, our experiments show LPS remains an effective approach to improve both shift consistency and model performance.

## 6 Conclusion

We propose learnable polyphase sampling (LPS), a pair of shift-equivariant down/upsampling layers. LPS's design theoretically guarantees circular shift-invariance and equivariance while being end-to-end trainable. Additionally, LPS retains superior consistency on standard shifts where theoretical assumptions are broken at image boundaries. Finally, LPS captures a richer family of shift-invariant/equivariant functions than APS. Through extensive experiments on image classification and semantic segmentation, we demonstrate that LPS is on-par with/exceeds APS, LPF and DDAC in terms of model performance and consistency.

**Acknowledgments:** We thank Greg Shakhnarovich & PALS at TTI-Chicago for the thoughtful discussions and computation resources. This work is supported in part by NSF under Grants 1718221, 2008387, 2045586, 2106825, MRI 1725729, NIFA award 2020-67021-32799, and funding by PPG Industries, Inc. We thank NVIDIA for providing a GPU.

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
