# OpenReview forum: "Learnable Polyphase Sampling for Shift Invariant and Equivariant Convolutional Networks"
_NeurIPS.cc/2022/Conference — NeurIPS 2022 Accept_

### Official Review · Reviewer_bbM7 · 2022-07-10

**Rating:** 6
**Confidence:** 3
**Soundness:** 3 good
**Presentation:** 3 good
**Contribution:** 3 good

**Summary:**

The paper tackles the issue of shift invariance/equivariance in CNNs. Prior work (APS) suggested, when downsampling, to select the index for whose value has the largest norm. This paper suggests a generalization which consists of learning the index-selection rule instead. Experiments on classification (invariance: CIFAR, ImageNet) and segmentation (equivariance ; PascalVOC) show convincing results.

Learning this selection is implemented via a carefully crafted small NN and the gumbel-softmax trick.

**Questions:**

See above.

**Limitations:**

None.

**Strengths And Weaknesses:**

## Strengths

- The paper clearly motivates the problem.
- The paper gives just the right amount of background knowledge and prior work.
- The evaluations are to the point.

## Weaknesses

The experiments show convincing results: shift invariance matching theory, and small but consistent improvement in "regular" accuracy measures. However, I wonder whether the small but consistent improvements may not be simply due to the fact of introducing more learnable parameters. This is almost addressed in Appendix A7, but what I am missing is the results of either (a) resnet + LPS but with slightly reduced ResNet size to match plain resnet params/speed, or (b) plain resnet with slightly increased size to match resnet + LPS params/speed.

Also, this should be mentioned in the main paper, with reference to the appendix. Actually, most of the appendix should be mentioned in the main paper.

---

> ### Author Response · Authors · 2022-08-01
> **Response to Reviewer bbM7**
>
> > ### Q1. Comparison with slightly increased model size to match resnet + LPS params/speed
>
> As summarized in Appendix Tab. A7, on ResNet101 + DDAC, our method increases the trainable parameters by 1.04%. Following the reviewer's suggestion, we tried modifying the ResNet101 + DDAC architecture by increasing its residual blocks (leading to 44,796,218 trainable parameters) to match our ResNet-101 + LPD model (44,751,034 trainable parameters). Specifically, we added three and one residual blocks to the first two ResNet stages. Despite the extra layers, we did not observe any gain in accuracy or shift-consistency over the original ResNet101 + DDAC model.
>
> We also compare our ResNet-101 + LPD model (44,751,034 trainable parameters)
> against an even larger ResNet-152 model (60,192,808 trainable parameters). Both
> models were trained on ImageNet using the same augmentation and optimizer
> configuration. ResNet-152 achieves 78.3% top-1 classification accuracy and 90.9%
> shift consistency, while our ResNet-101 + LPD model obtains 78.8% top-1
> accuracy and 92.4% shift-consistency. Despite having 25% less trainable
> parameters, our model attains 0.5% higher accuracy and 1.5% higher
> shift-consistency. These results show that the improved performance of our model
> is not due to its additional trainable parameters. We will update the appendix to include these results.
>
>
> > ### Q2. Reference to the appendix in the main paper
>
> Thanks for pointing this out. We did not reference the appendix to keep the paper self-contained. As suggested, we will add references to the appendix in the main paper.

---

> > ### Comment · Reviewer_bbM7 · 2022-08-08
> > **Thanks for your answers**
> >
> > Thanks for your answers, that extra experiment with the ResNet-152 is especially convincing, I recommend adding it to the appendix of the final paper.

---

> > > ### Author Response · Authors · 2022-08-09
> > > **Thanks for your feedback**
> > >
> > > Thank you for the time spent on reviewing and replying to our response. We are happy to see that the reviewer’s concerns have been resolved. As mentioned in our response, we will add these results to the appendix.

---

### Official Review · Reviewer_3D4j · 2022-07-11

**Rating:** 5
**Confidence:** 5
**Soundness:** 2 fair
**Presentation:** 3 good
**Contribution:** 2 fair

**Summary:**

This paper presents a  new framework to enable the design of shift invariant and equivalence convolutional neural networks. The key idea in this paper is introducing a  learnable polyphase sampling scheme which can be used in both down/up sampling layers.  Compared to existing methods, the proposed LPS is trained from data instead of manually.  The LPS is introduced with detailed  analysis and the implementation is also shown in both down/up sampling layers.  Experiments on image classification and semantic segmentation valid its effectiveness in practices.


**Questions:**

Q1.In general LPS has similar performance to APS in Resnet 18 setting. Is the result consistent in Resnet 50 or 101?


Q2.  Table 4 shows that  LPD only works better than LPS, while there is no ablation shows that the importance of LPD and LPU in the neural network when keep replacing LPD LPU in downsampling and upsampling layers.   Will is possible LPD + a few LPU layers  can have even better performance?



**Ethics Review Area:**

["Inadequate Data and Algorithm Evaluation"]

**Limitations:**

The limitation in this paper is the marginal performance improvement over APS, which leads the bias for justifying the importance of the motivation in this paper.   This is because APS is a special case of  LPS, the contribution of this paper can only be valid through theoretical analysis.

**Strengths And Weaknesses:**

Pros:
The main contribution in this paper is extending the manually designed adaptive polyphase sampling layers to learnable polyphase sampling layers with theoretically prove.  Especially, the introduced conditional probability directly learn from the data while the LPS keep the property of APS.   Besides the LPS, both downsampling and upsampling LPD and LPU have been showed with implementation and detailed analysis.

 Cons:
The main idea of this paper is generalized the existing work APS in a learnable manner.  Compared to the significance of APS the contribution is limited.  Also it is not clear how LPU and LPD work in real cases. For example, in Semantic Segmentation cases, LPD  only seems works better than LPD and LPU while it is not clear how LPU works and what the reason behind that.

---

> ### Author Response · Authors · 2022-08-01
> **Response to Reviewer 3D4j**
>
> We thank the reviewer for the feedback. We think there are some misunderstandings regarding the conducted experiments. Please see our responses below.
>
> > ### Q1. Contribution of Learnable Polyphase Sampling (LPS)
>
> We propose LPS, **a downsampling (LPD) and an upsampling (LPU) layer** which learn
> a truly shift-invariant downsampling scheme or a truly shift-equivariant
> upsampling scheme. This differs from the prior adaptive polyphase sampling (APS)
> method which **only considers downsampling** and which is based on a
> **hand-crafted rule**.  See Line 29 for a detailed discussion regarding their
> differences. Additionally, we believe there is value to generalizations. Our
> proposed method is more flexible than APS and we empirically demonstrate that
> LPS does not learn to recover APS (see Line 264). Lastly, our claims/proofs differ from APS’s:
>
> - (a) We introduce the shift-permutation property which describes how spatial shifts result in permutations of the polyphase indices (Lemma 1).
> - (b) Using Lemma 1, we prove a sufficient condition for constructing shift-equivariant downsampling layers (Claim 1).
> - (c) We prove that our proposed downsampling layer captures a more general family of downsampling layer than APS (Claim 2).
> - (d) We prove how to design practical models satisfying the sufficient condition in Claim 1 (Claim 3).
> - (e) We prove that the proposed down/upsampling (LDP/LPU) layers can be used to construct shift-equivariant deep-nets (Claim 4).
>
> **These results are novel and have not been considered by prior works.**
>
> > ### Q2. Effectiveness of Learnable Polyphase Upsampling (LPU) in practice...not clear how LPU works and what's the reason behind that.
>
> We study both a down- and upsampling layer (LPD/LPU). In the segmentation
> experiments, LPS refers to using both LPD **and** LPU. As can be seen in Tab. 4,
> we performed an ablation study with **LPD only**. This baseline uses standard
> bilinear interpolation for upsampling following DeepLabV3+ and is not truly
> shift equivariant. In contrast, LPS achieves 100% circular shift consistency while
> maintaining model accuracy; this highlights the benefit of having LPU
> layers.
> The description of LPU is defined in Sec. 4.3 (Line 187), where upsampling is performed by placing
> features back to the original spatial location prior to downsampling. The other spatial locations are filled with zeros,
> as summarized in Eq. (24). We also theoretically prove that the proposed LPD+LPU (LPS) together results in truly shift-equivariant deep-nets (see Claim 4).
>
> > ### Q3. In general, LPS has similar performance to APS in a ResNet 18 setting. Is the result consistent in ResNet 50 or 101?
>
> As the APS paper did not report ResNet50/101 results for ImageNet, we only
> compared to the more recent SOTA method: DDAC. To answer the question, we ran additional
> experiments on ResNet-101 (ImageNet), comparing APS and our proposed LPS method.
>
> The results show, both methods attain
> comparable top-1 classification accuracy, while our proposed LPS improves shift
> consistency (see results below). We will add these results to Tab. 3 of the
> main paper.
>
> | Method | Anti-Alias | Acc. (%) | S-Cons. (%) |
> |------------|------------|----------|-------------|
> | LPF | Tri-3 | 78.4 | 91.6 |
> | DDAC | Adapt-3	| 79.0 | 91.8 |
> | DDAC* | Adapt-3 | 78.6 | 91.8 |
> | APS      | Adapt-3	| 78.8 	| 92.0|
> | LPS (ours) | Adapt-3	| 78.8 	| 92.4 |
>
>
> > ### Q4. Table 4 shows that LPD only works better than LPS. Will LPD + a few LPU layers have even better performance?
>
> This is a misunderstanding. LPS refers to using both LPD/LPU layers, i.e., the architecture uses LPD for downsampling and LPU for upsampling (as requested). In Line 29, we explained LPS to be “**a pair of down/upsampling layers**”. We will clarify this in the experiment section. As can be seen in Tab. 4, LPS achieves perfect circular shift consistency with competitive model performance.

---

> > ### Comment · Reviewer_3D4j · 2022-08-09
> > **Further questions**
> >
> > Thanks for putting all the explanation and resolving my concerns.  I notice that for LPS, in terms of Acc and Cons metrics, it is very hard to improve both at same time.    I am curious about any ablation studies  on # of LPS layers for the results? Will it possible to find sweet spot on improving both metric?  Any comments on it? No need to do such experiments for my answer.

---

> > > ### Author Response · Authors · 2022-08-09
> > > **Re: Further questions**
> > >
> > > Interesting thought. In our classification experiments, we replaced **all** downsampling layers with LPD. The number of LPD layers is hence determined by the architecture. In our segmentation experiments, we replaced **all** down/upsampling layers with LPD/LPU respectively. We have not experimented with replacing only some of the down/up- sampling layers with LPD/LPU.
> > >
> > > Looking at Table 3, we agree with the reviewer that there is likely a sweet spot, e.g., 2 LPD followed by 2 standard downsampling layers. Anecdotally, we think it is more crucial for the earlier layers to be shift-equivariant. However, we need to conduct additional experiments for a conclusive claim.

---

### Official Review · Reviewer_MjxQ · 2022-07-15

**Rating:** 7
**Confidence:** 3
**Soundness:** 3 good
**Presentation:** 3 good
**Contribution:** 3 good

**Summary:**

This paper generalizes polyphase resampling and applies learnable selection of the polyphase components, instead of based on signal magnitudes. It achieved better accuracy and shift consistency on classification and segmentation.

**Questions:**

Maybe I missed something, as APS is a specicial case of LPU, and why is LPU more shift-invariant than APS theoretically, and on segmentation empirically? As LPU is data dependent, so it could be approximate APS on some bad test case?

In figure 4 caption "This indicates LPS did not learn to reproduce APS" I can infer this from the algorithm, but is this good from visual meaning perspective? Difficult to see.

The running time, computation complexity comparison with APS, and other baseline should be included.

**Limitations:**

yes

**Strengths And Weaknesses:**

Strength: The paper writes with professionism in notations and arangement. The authors did both experiment in high-level classification and low-level segmentation.

Weakness: My personal feeling here is that: the idea of polyphase resampling is being derived to maybe too complicated, to lose its original meaning in signal processing, and now it becomes a fancy way of 2x2 pooling. Even APS can be used for backpropagation actually, it is just another non-smooth operator adding to some existing non-smooth operators, Relu, maxpooling, so no big trouble. So I am conservative about the motivation to be the polyphase resampling parameterized. The computation added by this layer cannot be neglet, considering the shiting doubles of workload of whatever layer follows it.

---

> ### Author Response · Authors · 2022-08-01
> **Response to Reviewer MjxQ**
>
> We thank the reviewer for appreciating our paper’s writing and professionalism.
>
> > ### Q1. Motivation of learnable polyphase sampling (LPS) and its naming
>
> In this paper, we propose LPS, a pair of down/up-sampling (LPD/LPU) layers which
> **learn** a truly shift-invariant/equivariant down/up sampling scheme. This
> differs from the prior adaptive polyphase sampling (APS) method which (a) only considers
> downsampling, and (b) is based on a “hand-crafted” selection rule. For consistency,
> we followed their choice of using the term polyphase sampling. We are happy to
> take naming suggestions.
>
>
> > ### Q2. Computation added by this layer
>
> In the appendix Tab. A7, we report the inference time for each of the models. On ResNet-101 with DDAC, our method increases the inference time by roughly 4.81%. Please see Appendix A4.3 for more details on inference time and architectures.
>
>
> > ### Q3. APS is a special case of LPS
>
> Sorry for the confusion. APS is only a downsampling layer, hence it is a special case of LPD. In theory, both APS and LPD achieve 100% circular shift consistency (Claim 1). Different from APS, LPD represents a larger family of downsampling selection schemes, which broadens the type of function a deep-net can represent. Specifically, Claim 2 states “APS is a special case of LPD, i.e., LPD can represent APS’s selection rule”.
>
> On segmentation, our proposed method (LPS=LPD+LPU) achieves 100% circular shift consistency
> both in theory and in practice (Claim 4). In contrast, APS is **only a downsampling
> scheme**. Therefore, this APS baseline uses standard bilinear interpolation for
> upsampling following DeepLabV3+ and is not guaranteed to be circularly shift
> equivariant. We will clarify this detail in the paper.
>
>
> > ### Q4. Visual meaning of Figure 4
>
> Fig. 4 is meant to qualitatively show that LPD learns to select different
> components during downsampling from APS. We did not claim the visualized feature spaces are interpretable. At
> Line 264, we explicitly stated: “we did not observe a specific pattern that can
> explain LPD’s selection rule”. We think interpretable visualization of the
> feature space is itself an interesting research topic.
>
>
> > ### Q5. Running time, computation complexity should be included
>
> The running time and introduced trainable parameters of our method are documented in the appendix. Please see Appendix A4.3 and Tab. A7 for a detailed comparison between methods. We will include this section in the main paper.

---

### Meta-Review · Area_Chair_6RRB · 2022-08-27

**Recommendation:** Accept
**Confidence:** Less certain

**Metareview:**

The paper proposes an end-to-end learnable polyphase sampling and shows competitive performance and sufficient novelty. Major concerns of the reviewers seem to be addressed during the rebuttal and therefore it can be accepted.

**Award:**

No

---

### Decision · Program_Chairs · 2022-09-14

Accept